# Dynamics of nevus development implicate cell cooperation in the growth arrest of transformed melanocytes

Rolando Ruiz-Vega[1,2], Chi-Fen Chen[3], Emaad Razzak[1], Priya Vasudeva[3], Tatiana B Krasieva[4], Jessica Shiu[3], Michael G Caldwell[1], Huaming Yan[5], John Lowengrub[1,5], Anand K Ganesan[1,3], Arthur D Lander[1,2,6]*

[1]Center for Complex Biological Systems, University of California, Irvine, Irvine, United States; [2]Department of Developmental and Cell Biology, University of California, Irvine, Irvine, United States; [3]Department of Dermatology, University of California, Irvine, Irvine, United States; [4]Beckman Laser Institute, University of California, Irvine, Irvine, United States; [5]Department of Mathematics, University of California, Irvine, Irvine, United States; [6]Department of Biological Chemistry, University of California, Irvine, Irvine, United States

*For correspondence:
adlander@uci.edu

Competing interests: The authors declare that no competing interests exist.

**Abstract** Mutational activation of the *BRAF* proto-oncogene in melanocytes reliably produces benign nevi (pigmented 'moles'), yet the same change is the most common driver mutation in melanoma. The reason nevi stop growing, and do not progress to melanoma, is widely attributed to a cell-autonomous process of 'oncogene-induced senescence'. Using a mouse model of Braf-driven nevus formation, analyzing both proliferative dynamics and single-cell gene expression, we found no evidence that nevus cells are senescent, either compared with other skin cells, or other melanocytes. We also found that nevus size distributions could not be fit by any simple cell-autonomous model of growth arrest, yet were easily fit by models based on collective cell behavior, for example in which arresting cells release an arrest-promoting factor. We suggest that nevus growth arrest is more likely related to the cell interactions that mediate size control in normal tissues, than to any cell-autonomous, 'oncogene-induced' program of senescence.

## Introduction

Activating BRAF mutations (e.g. BRAF^V600E) are the most common oncogenic mutations in melanoma, seen in about 66% of cases (*Davies et al., 2002*). Curiously, the same mutation is found in 89% of melanocytic nevi (*Pollock et al., 2003*)—the benign, pigmented 'moles' found on the skin of most individuals. In animal studies, melanocyte-specific expression of BRAF^V600E efficiently produces nevi, but only very rarely melanoma (*Dankort et al., 2009*; *Dhomen et al., 2009*; *Patton et al., 2005*). The widely-accepted explanation is that transformed melanocytes undergo oncogene-induced senescence (OIS), arresting proliferation before additional oncogenic events can occur (e.g. *Bennett, 2003*; *Huang et al., 2017*; *Kaplon et al., 2014*; *Michaloglou et al., 2005*).

Nevus melanocytes are indeed growth-arrested, but the assumption that OIS is the cause remains untested, in part because of a lack of criteria to rigorously define OIS *in vivo* (*Damsky and Bosenberg, 2017*). Initially studied as a consequence of forced expression of oncogenes in cell cultures (*Serrano et al., 1997*), OIS has come to be seen as a distinctive cellular stress response characterized by a phenotype of growth arrest, morphological and metabolic changes, chromatin alterations, and secretion of growth factors, chemokines, cytokines and proteases (*Campisi and d'Adda di Fagagna, 2007*; *Gorgoulis et al., 2019*; *Ito et al., 2017*; *Kuilman et al., 2010*).

**eLife digest** Melanocytes are pigment-producing cells found throughout the skin. Mutations that activate a gene called *BRAF* cause these cells to divide and produce melanocytic nevi, also known as "moles". These mutations are oncogenic, meaning they can cause cancer. Indeed, *BRAF* is the most commonly mutated gene in melanoma, a deadly skin cancer that arises from melanocytes. Yet, moles hardly ever progress to melanoma.

A proposed explanation for this behavior is that, once activated, *BRAF* initiates a process called "oncogene-induced senescence" in each melanocyte. This process, likened to premature aging, is thought to be what causes cells in a mole to quit dividing. Although this hypothesis is widely accepted, it has proved difficult to test directly.

To investigate this notion, Ruiz-Vega et al. studied mice with hundreds of moles created by the same *BRAF* mutation found in human moles. Analyzing the activity of genes in individual cells revealed that nevus melanocytes that have stopped growing are no more senescent than other skin cells, including non-mole melanocytes.

Ruiz-Vega et al. then analyzed the sizes at which moles stopped growing, estimating the number of cells in each mole. The data were then compared with the results of a simulation and mathematical modeling. This revealed that any model based on the idea of cells independently shutting down after a number of random events could not reproduce the distribution of mole sizes that had been experimentally observed. On the other hand, models based on melanocytes acting collectively to shut down each other's growth fit the observed data much better.

These findings suggest that moles do not stop growing as a direct result of the activation of *BRAF*, but because they sense and respond to their own overgrowth. The same kind of collective sensing is observed in normal tissues that maintain a constant size. Discovering that melanocytes do this not only sheds light on why moles stop growing, it could also help researchers devise new ways to prevent melanomas from forming.

Given an abundance of 'hallmarks' of senescence, one might think that recognizing this cell state in vivo should be straightforward. Yet no single hallmark distinguishes senescence from other growth-arrested cell states. Phenotypes once thought to be 'gold standards', such as expression of lysosomal beta-galactosidase, cyclin-dependent kinase inhibitors, or p53, commonly mark only subsets of senescent cells (*Wiley et al., 2017*), as well as non-senescent cells (*Tran et al., 2012*). Moreover, observations of supposedly senescent cells resuming proliferation (e.g. *Beauséjour et al., 2003*), imply that permanent cell cycle exit cannot be used as a distinguishing feature. *In vivo* senescence, as a result, is currently somewhat of a *Gestalt* diagnosis, that is assessed by a collection of traits, no subset of which is necessary or sufficient. Yet there is no clear consensus on which traits are best to assess, and recent meta-analyses of gene expression suggest that some of the most commonly assessed features are not 'core' to senescence at all (*Hernandez-Segura et al., 2017*).

The reason it is important to clarify how BRAF-transformed nevus melanocytes stop growing is that it shapes how we think about the origins of melanoma. OIS is usually portrayed in *cell-intrinsic* terms: oncogene expression within a transformed cell produces a stress within *that* cell, which triggers *it* to senesce. Even those who acknowledge a possible role for paracrine signals (*Acosta et al., 2013*; *Elzi et al., 2012*; *Ito et al., 2017*; *Wajapeyee et al., 2008*) still portray the process as something initiated and orchestrated by cell-autonomous responses to oncogenes. This naturally leads to an approach to melanoma prevention and treatment that focuses on understanding how oncogenes derange intracellular processes; how those derangements elicit stress responses; and what might enable cancer cells to circumvent those responses (e.g. *Bennett, 2003*; *Damsky and Bosenberg, 2017*; *Vredeveld et al., 2012*; *Yu et al., 2018*). In contrast, as we argue below, it is possible that the growth arrest displayed by nevus melanocytes has little to do with oncogene-induced stress, and may have more to do with networks of cell–cell communication that are characteristic of melanocytes, independent of whether they are transformed. In this case, the most effective path to understanding how to prevent or treat melanoma could be to better elucidate the normal physiology of melanocytes in their environment.

Here, we investigate the details of nevus growth arrest in a model in which melanocyte-specific *Braf* activation generates hundreds of nevi on the skin of mice (*Dankort et al., 2009*). By examining both single-cell transcriptomes and the dynamics of growth arrest in nevus-associated melanocytes, we make two key observations: First, patterns of gene expression in arrested nevus melanocytes fail to identify them as any more senescent than other skin cells or normal melanocytes, arguing against a primary role for any form of senescence in their arrest. Second, the timing and statistics of nevus formation effectively argue against *any* relatively simple cell-autonomous process as being the cause of growth arrest. Ultimately, we propose a model in which arrest is driven not by oncogene stress, but by feedback mechanisms similar to those commonly involved in normal tissue homeostasis.

## Results

### Dynamics of nevus growth

Characterizing the dynamics of nevus growth and arrest requires observing nevi that started growing at known times. We took advantage of a mouse model in which Cre-mediated recombination introduces the activating V600E mutation into the endogenous *Braf* locus. When crossed onto a background carrying a *Tyr-CreER* transgene, the mice acquire the $Braf^{V600E}$ mutation only in cells of the melanocytic lineage, and only after Cre activation by 4-hydroxytamoxifen (4-OHT), applied either systemically or through painting on the skin.

As shown previously (*Dankort et al., 2009*), 4-OHT treatment of these mice leads to development of numerous pigmented nevi. Visualization of nevi is hindered, however, by the strong pigmentation in hair follicles which, except at microscopic resolution, can be difficult to distinguish from nevi. One way to circumvent this difficulty is to observe nevi only during the telogen phase of the hair cycle, when follicle-associated pigment is not present (conveniently, synchronization of hair cycles may be maintained on a large patch of skin through depilation).

As shown in *Figure 1*, in mice whose back skin was treated with topical 4-OHT at postnatal day 2 (P2), P3 and P4, nevi were apparent macroscopically at telogen (P50; *Figure 1A*). Live imaging, using multi-photon microscopy (MPM; *Saager et al., 2015*), revealed that, like human nevi, mouse nevi consist of scattered nests of pigment-containing cells (*Figure 1B*). Nevi could also be visualized post-mortem, using a dissecting microscope, on the undersurface of pieces of telogen-stage back skin (*Figure 1C*).

An alternate approach to visualization that did not require hair synchronization was to generate nevi by painting 4-OHT on glabrous (hairless) skin, such as the ventral surface of the paw, permitting tracking of individual nevi on a daily basis. As shown in *Figure 1D*, when forepaws were treated with 4-OHT from P2 through P4, tiny nevi could be detected as early as P6. Serial observation indicated that most nevi reach a maximum size somewhere between P16 and P21 (*Figure 1D* , *Figure 1—figure supplement 1A*). This suggests that, in the mouse, $Braf^{V600E}$-transformed melanocytes arrest within 2–3 weeks. To confirm this, we used BrdU labeling to monitor DNA synthesis. Because melanin readily obscures immunohistochemical signals, these experiments were done in an albino (unpigmented) genetic background, using premelanosome protein (Pmel) staining to identify melanocytes. As shown in *Figure 1F*, albino mice generate nests similar to those seen in pigmented mice. In such animals, BrdU readily incorporated into hair follicle melanocytes (*Figure 1E*, *Figure 1—figure supplement 1B*), whereas by p21 nests within nevi were uniformly negative for BrdU, implying growth arrest (*Figure 1F*, *Figure 1—figure supplement 1C*).

The conclusion that Braf-induced nevi are already growth-arrested by P21 agrees with the reports of others (*Damsky and Bosenberg, 2017*), and is lent further support by time course measurements of nest size by MPM (*Figure 1C*), which show that nest size distributions change insignificantly between P21 and P50 (*Figure 4—figure supplement 1A-B*).

### Do nevi undergo OIS?

As discussed above, senescence is usually accompanied by distinctive gene expression. Various gene expression 'signatures' have been developed to help investigators identify senescent cells and distinguish them from cells that have become growth-arrested by other processes. We considered several of these (*Source data 1*):

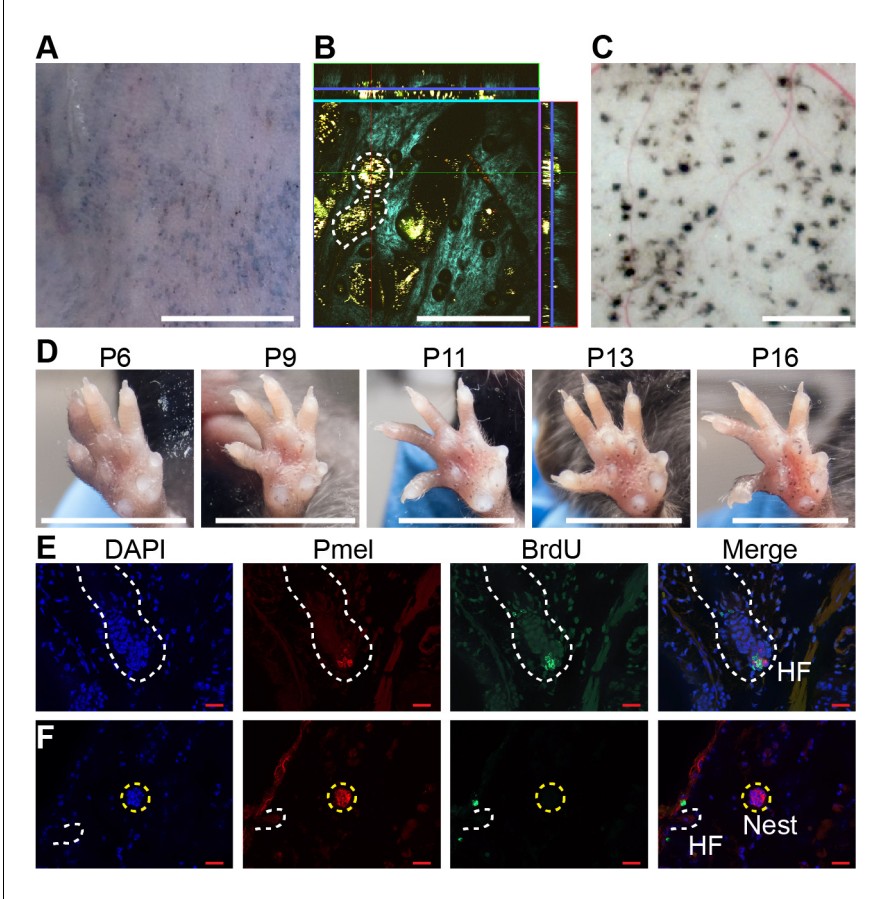

**Figure 1.** Dynamics of nevus growth.  (A-D) Visualization of nevi on Braf^V600E mice. (A) Live imaging of back skin at telogen-stage (P50), following hair depilation. Scale bar = 5 mm. (B) Live imaging of a sample like that in panel A using multi-photon microscopy. The central square is an *en face* view of the skin (*x-y* plane), while rectangles above and to the right are cross-sections (*x-z* and *y-z* planes, respectively, with blue lines marking the location of the central image). Melanin autofluorescence appears yellow, second harmonic generation of collagen is cyan, and keratin autofluorescence is green. Dashing outlines dermal melanocyte nests. Scale bar = 318 µm. (C) Appearance of nevi on the undersurface of back skin (at P21). Scale bar = 1 mm. (D) Nevus development on the ventral (glabrous) surface of the paw. Images of a single paw were taken at the indicated ages. Scale bar = 0.5 cm. (E-F) Assessment of melanocyte proliferation. Sections are from albino wildtype (E) and Braf^V600E (F) skin at P21. Melanocytes were identified by premelanosome protein (Pmel) immunohistochemistry and proliferation assessed by BrdU incorporation. Wildtype hair follicle (HF) melanocytes (E) incorporate BrdU whereas nevus melanocytes (F) do not. Scale bar = 20 µm.

The online version of this article includes the following figure supplement(s) for figure 1:

**Figure supplement 1 .** Dynamics of nevus growth.

1. A set of genes encoding the most commonly considered 'hallmarks' of senescence, that is p53, Rb, lysosomal beta-galactosidase, H2AX, and three cyclin-dependent-kinase inhibitors ('Classical', seven genes [*Campisi and d'Adda di Fagagna, 2007*; *Collado and Serrano, 2006*]);
2. A gene signature that distinguishes cultured human fibroblasts growth-arrested by BRAF-transformation from quiescent fibroblasts ('Kuilman', 21 genes [*Kuilman et al., 2008*])
3. Results of a meta-analysis (*Hernandez-Segura et al., 2017*) of publications on fibroblasts, melanocytes, and astrocytes, comparing senescence (induced by multiple different stresses) with quiescence, yielding 'universal' signatures of genes that are up- and downregulated specifically in senescence ('Universal Up', 31 genes, and 'Universal Down', 23 genes) as well as signatures of genes specifically up- or down-regulated by senescence induced in melanocytes ('Melanocyte Up', 397 genes and 'Melanocyte Down', 135 genes).

4. The most statistically significant genes in a recent meta-analysis (*Chatsirisupachai et al., 2019*) of 20 replicative senescence microarray datasets from the Gene Expression Omnibus ('Chatsiri-supachai Up', 237 genes and 'Chatsirisupachai Down', 244 genes).

5. A list of genes characteristic of the 'Senescence-Associated Secretory Phenotype' ('SASP', 81 genes), compiled from 38 literature references (for citations see *Source data 1*).

To determine whether any of these proposed signatures fits nevus melanocytes, we performed single-cell RNA sequencing on dissociated cells from the back skin of nevus-bearing mice at both P30 and P50 (i.e. after nevi have stopped growing), using wildtype skin as a control. Using known cell-type marker genes (*Figure 2—figure supplement 1A-B*), we identified 14 different cell types in the skin, including melanocytes (*Figure 2A*). Unsupervised clustering further sub-divided the melanocytes into four groups (*Figure 2B*): Two of them, Mel 0 and Mel 1, were composed of cells found only in nevus-bearing, and not wildtype, skin (*Figure 2C*); they are highly similar in gene expression, primarily differing in having a slightly lower level of pigment gene expression in Mel 1 versus Mel 0 (*Figure 2D*). We identify them as the 'nevus melanocytes', because they are seen only when nevi are present, and are by far the predominant melanocyte population in such animals.

Mel 2 cells express the lowest levels of pigmentation genes (*Figure 2D*), and are seen in both genotypes (*Figure 2C*) at all stages (although expanded in number in nevus-bearing animals (*Figure 2F*)). Their pattern of gene expression bears a strong resemblance to one recently published for melanocyte stem cells isolated from telogen-stage hair follicles (*Zhang et al., 2020*). In particular, they express *Cd34*, which has been proposed to be a marker for bulge-associated melanocyte stem cells (*Joshi et al., 2019*).

Finally, cells of cluster Mel 3, which express the highest levels of pigment genes (*Figure 2D*), are found in both mutant and wildtype mice, but only at the P30 time point (*Figure 2E–F*). We thus identify them as mature hair follicle melanocytes, as such cells are present exclusively during anagen phase of the hair cycle (P30), and disappear during telogen (P50).

Because gene signatures are based on the idea of up- and downregulation of expression relative to some baseline state, to test whether nevus melanocytes fit a known signature it is necessary to have comparison transcriptomes. We made two types of comparisons: nevus melanocytes versus every other cell type in the skin (which, with the possible exception of mature keratinocytes, we would not expect to be senescent); and the four melanocyte subclusters (two of which are nevus-associated and two of which are not) versus each other. In each case we computed average expression for each gene in every cell type or cluster, together with a standard error of the mean as a measure of dispersion. Expression values were then normalized to average expression across all of the cells being compared (i.e. all skin cells, or all melanocytes, depending on which comparison was being done) and $log_2$-transformed, so that positive values signify upregulation (relative to the average for that gene), and negative downregulation. Gene expression was then visualized using heat maps (*Figure 3A* and *Figure 3—figure supplements 1–2*, with positive values in blue and negative in red). Because gene expression levels inferred from single-cell RNA sequencing tend to be noisy, particularly for genes with low expression, we ranked all genes by their minimum level of noise (i.e. normalized standard error of the mean in the least noisy cell type), and used this value ('n-SEM', which is also presented graphically as a bar to the right of each heatmap) to sort gene lists, so that maps vary from most to least reliable as one goes from top to bottom.

*Figure 3A* shows the results for the 'Classical' and 'Universal Up' signatures (heat maps for the other signatures are shown in *Figure 3—figure supplement 1*). Here we see no strong enrichment of blue over red signals in nevus melanocytes, nor in most other cells. When compared with whole skin, using the 'Classical' signature, only *Cdkn2a* stands out as strongly upregulated in nevus melanocytes, but it is similarly upregulated in skin fibroblasts (it also has the noisiest data among genes in the signature). With the 'Universal Up' signature, more genes are downregulated than upregulated in nevus melanocytes. To quantify such impressions, we summed the $log_2$-transformed data in each column in every heat map, producing the bar graphs in *Figure 3B*. We reasoned that summation of log-transformed data would emphasize consistent trends in the data while suppressing effects of noise (random positive and negative variation would tend to cancel out). The results suggest that skin fibroblasts better fit the 'Classical' senescent signature than any skin cell type, including nevus melanocytes, or indeed melanocytes of any cluster. As a control—to demonstrate the ability of this approach to correctly associate cell types with gene signatures— we analyzed the same data using a signature of cell proliferation, 'meta-PCNA', that represents 129 human genes

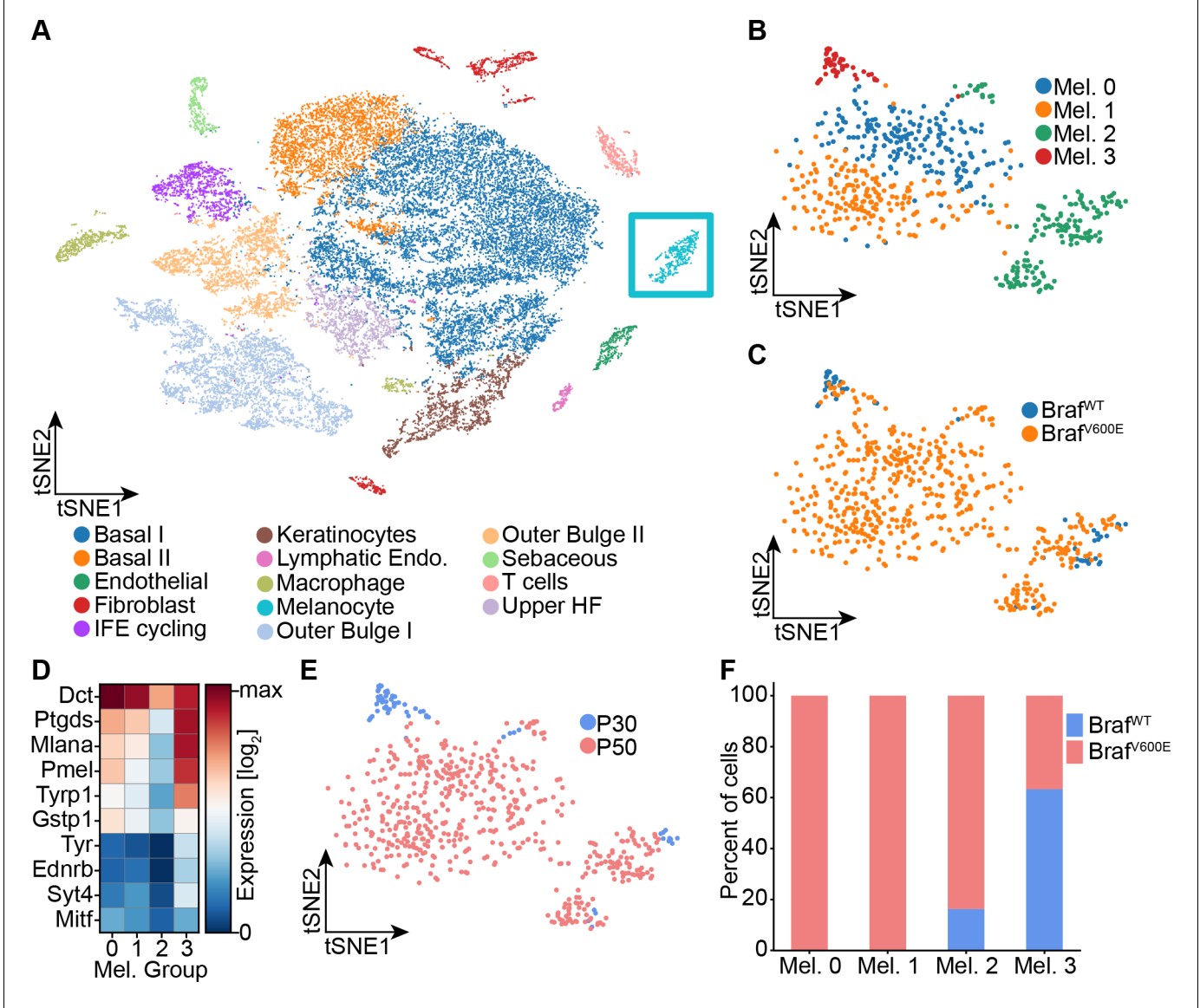

**Figure 2.** Single-cell RNA sequencing of mouse dorsal skin to transcriptionally characterize melanocytes. (A) Skin cell types are visualized with tSNE (cells = 35,141) from mice at P30 (n[Braf$^{WT}$]=2 mice, n[Braf$^{V600E}$]=2 mice) and P50 (n[Braf$^{WT}$]=3 mice, n[Braf$^{V600E}$]=3 mice). Melanocytes are outlined with a blue box. (B) Subclustering of melanocytes (n = 609) visualized on a tSNE plot. Four clusters were identified. (C) Visualization of melanocytes based on their genotype on a tSNE plot. (D) A heat map of genes involved in pigmentation. Each cluster expresses these genes at different levels. (E) Visualization of melanocytes, based on mouse age, on a tSNE plot. (F) Quantification of melanocytes in each cluster based on their genotype (*BRAF* wildtype or mutant).

The online version of this article includes the following figure supplement(s) for figure 2:

**Figure supplement 1.** Single cell RNA sequencing of mouse dorsal skin, to transcriptionally characterize cells.

most positively correlated with proliferation marker PCNA in a compendium of normal tissues (*Venet et al., 2011*). As shown in *Figure 3B* (also see *Figure 3—figure supplement 1*), this signature (122 genes of which had unambiguous mouse orthologs; *Source data 1*) identified two keratinocyte populations ('IFE-cycling' and 'Outer Bulge 1') as highly proliferative (in agreement with *Joost et al., 2020*), and mature (postmitotic) keratinocytes as non-proliferative. Importantly, it also correctly identified nevus melanocytes as non-proliferative—and other melanocytes as proliferative—in agreement with *Figure 1E–F*. Interestingly, the relatively high expression of proliferation-associated genes in non-nevus, hair follicle melanocytes (Mel 3) when compared with nevus

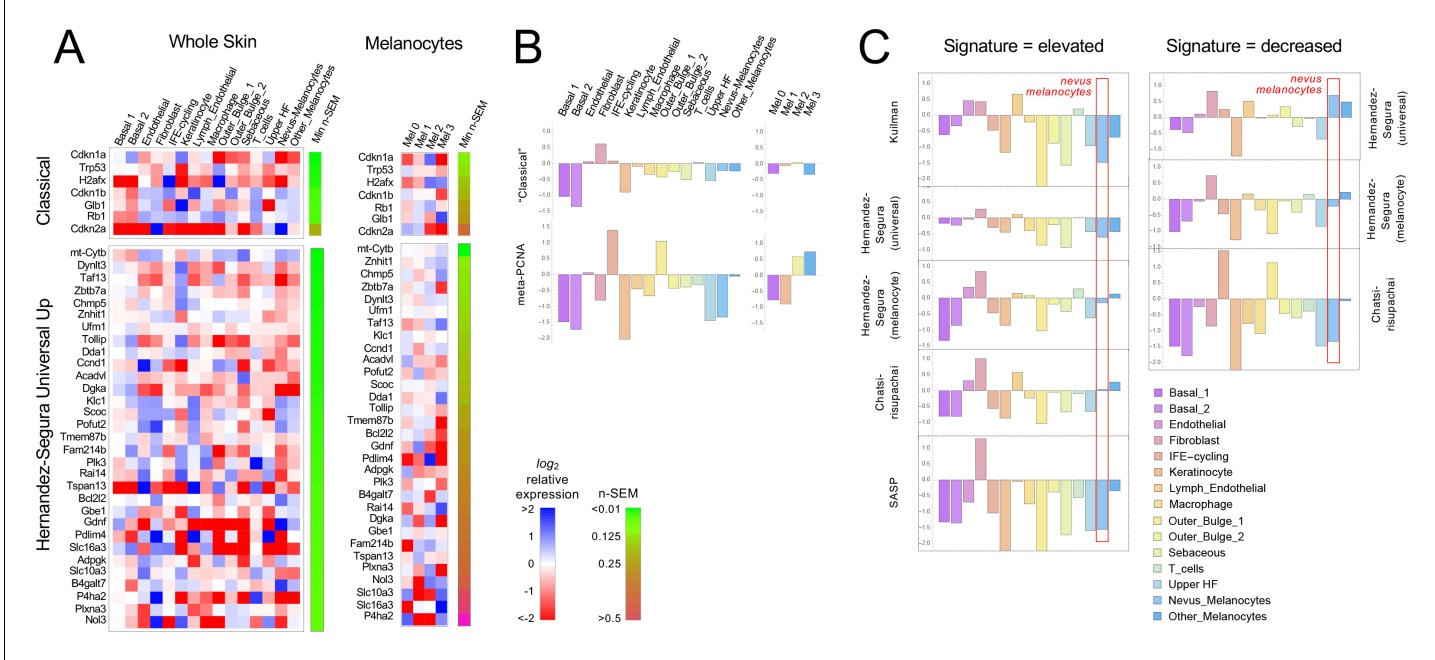

**Figure 3.** Gene expression fails to identify nevus melanocytes as senescent. Transcriptomes of clusters identified in *Figure 2* were compared with proposed 'signatures' of senescence.( **A**) Gene expression data for clusters in *Figure 2A and B* were averaged by cluster and, for each gene, expressed as a ratio to the average expression level of that gene in the entire skin sample ('Whole Skin') or just the melanocyte clusters ('Melanocytes'). The heat map displays the logarithm of that ratio, with blue representing upregulation and red downregulation, relative to the appropriate average. Two proposed signatures for genes upregulated in senescence ('Classical' and 'Universal Up') are shown (For other signatures, see *Figure 3—figure supplement 1*). Gene lists have been sorted by the minimum gene expression variability within the least variable cluster (green-brown bar). (**B**). The 'Classical' heatmaps in panel A are summarized as a bar graph displaying the sum of the log-transformed data. Also shown are summary results for a proposed signature of proliferation, 'meta-PCNA', which clearly distinguishes between cell types expected to be proliferative and non-proliferative (in normal skin). The 'Classical' senescence signature fails to identify nevus melanocytes (Mel 0 and Mel 1) as senescent, especially when compared with other cell types or other melanocytes. (**C**). Extension of the analysis in panel B to eight additional signatures.

The online version of this article includes the following source data and figure supplement(s) for figure 3:

**Figure supplement 1.** Heat maps for other signatures associated with senescence or proliferation.

**Figure supplement 2.** Comparing proliferation-associated gene expression in cluster 3 (hair follicle) melanocytes from wildtype and *Braf*-mutant animals.

**Figure supplement 3.** Comparing mouse nevus melanocyte gene expression with effects of BRAF$^{V600E}$ on cultured human melanocytes.

**Figure supplement 3—source data 1.** Data are derived from Table S1 of *Pawlikowski et al., 2013*, and relate to *Figure 3—figure supplement 3*.

melanocytes, was consistent between Braf$^{V600E}$-expressing and control mice (*Figure 3—figure supplement 2*), suggesting that tissue context plays a role in whether Braf$^{V600E}$-expressing cells even arrest growth.

*Figure 3C* and *Figure 3—figure supplement 1* extend this analysis to the remaining eight potential signatures of senescence (five consisting of genes that are upregulated; three of genes that are downregulated). In five of the eight cases, nevus melanocytes rank as *less* senescent than the average skin cell; in two of the cases nevus melanocytes are about average. In only one case ('Chatsirisupachai Down') does nevus melanocyte gene expression go in the predicted direction for senescence. However, the Chatsirisupachai signatures had not been curated to remove genes associated simply with proliferation/quiescence (*Chatsirisupachai et al., 2019*), and inspection of the 'Chatsirisupachai Down' gene list shows that 61 of its 250 genes are shared with the 129-gene meta-PCNA signature; that is it is more likely a signature of proliferation than 'non-senescence' (note the strong similarity between the 'Chatsirisupachai Down' bar graph in *Figure 3C* and the meta-PCNA graph in *Figure 3B*).

To confirm that the senescence-associated gene expression signatures used here truly could identify melanocytes that had become senescent, we also analyzed published data comparing gene expression in *BRAF$^{V600E}$*-transduced and normal human melanocytes in culture, under conditions in

which the former developed definitive morphological characteristics of senescence (*Pawlikowski et al., 2013*; see *Figure 3—figure supplement 3*). Of the 23 'Universal Down' signature genes, 15 were significantly differentially expressed, and 100% of these were decreased in expression. Of the 31 'Universal Up' genes, 19 were significantly differentially expressed, and 84% of these were elevated in expression.

Together these data do not support the view that any sort of senescence—oncogene-induced or otherwise—is characteristic of nevus melanocytes and therefore a possible cause of their growth arrest.

## Does a cell-autonomous process arrest nevi?

As discussed above, OIS is usually presented as a cell-autonomous process (e.g. *Dankort et al., 2009*; *Dhomen et al., 2009*; *Michaloglou et al., 2005*; *Serrano et al., 1997*). The simplest cell-autonomous process that one might imagine is a probabilistic switch: Once oncogene activation commences, cells arrest with a fixed probability (per time or per cell cycle). Regardless of the molecular details, such a model makes distinctive predictions about clonal dynamics.

Consider the clonal descendants of a single oncogene-transformed founder cell. For any value of the per-cell-cycle senescence probability (which we denote here as '*s*'), how many cells should we expect that clone to contain at any given time? How many cell cycles should it take before all of the cells in most clones should have arrested? Such questions are well studied in mathematics (*Athreya and Ney, 1972*), and easily solved by computer simulation. For this particular problem, there are two key results (*Figure 4*).

First, the time after which one can expect clones to have stopped growing (e.g. when all cells will have arrested in, say, 95% or 99% of clones) is a steep function of *s*. If *s* < 0.5, (i.e. less than a 50% chance of arrest per cell cycle), then some clones will never stop growing. If *s* is, say, 0.53, all clones will eventually stop growing, but one must wait 51 cell cycles before 99% of them do so (*Figure 4A*). Given typical lengths of postnatal mammalian cell cycles, and the fact that we observe cessation of mouse nevus growth in about 2–3 weeks, we may consider 30 to be a generous estimate for the maximum number of cell cycles by which nevi stop growing. To achieve 99% clonal arrest by 30 cell cycles, *s* must be around 0.56 or higher; to achieve arrest in 95% of clones, *s* must be greater than 0.52 (*Figure 4A*).

From the same simulations one may also calculate predicted distributions of clone sizes. There is a clear reciprocal relationship between mean clone size and the fraction of clones that arrest by 30 cell cycles of time (*Figure 4B*). For example, a value of *s* that enables 95% of clones to arrest produces a mean clone size of only 18.5 cells. For comparison, we estimate cell numbers per nevus to be in the range of 100–1000 cells (see Materials and methods).

The explanation for the small mean clone sizes produced by simulations can be appreciated by examining the full size distributions. As shown in *Figure 4C–D*, such distributions are extremely heavy-tailed, with a very large number of very small clones and a small number of very large clones (the histograms in *Figure 4C–D* are plotted with logarithmic abscissa to facilitate display of all clone sizes).

Qualitatively, this is very different from what we observed for nevi on the backs of p21 mice. Nevi displayed a mean radius of 76.8 μm (corresponding to an area of 0.019 mm$^2$, in excellent agreement with the results of *Damsky et al., 2015*) and, when plotted on a logarithmic scale, individual radii displayed a Gaussian-like distribution (*Figure 4E*; a Gaussian shape on a logarithmic axis is usually referred to as 'log-normal'). Interestingly, nest sizes (quantified by MPM) also seem to be log-normally distributed (*Figure 4—figure supplement 1*) whether at P21 (panel A) or P50 (panel B). So are the nests within human melanocytic nevi, despite the latter being an order of magnitude larger than those in mice (*Figure 4—figure supplement 1C*). It should be noted that using different units to represent simulation results (cell numbers) and empirical data (linear dimension) in *Figure 4* and its supplement does not confound comparing the shapes of distributions, thanks to the logarithmic abscissa: as long as cell number scales as some power of linear dimension, values associated with log-transformed bins are simply scaled by a constant factor.

The above comparison of observed distributions with the results of computer simulation is not entirely fair, however: Simulations track all clones, no matter how small, whereas empirical distributions undoubtedly omit nevi with sizes below some threshold of observability. To correct for this, one can truncate simulated distributions to remove clones smaller than some threshold size. With no

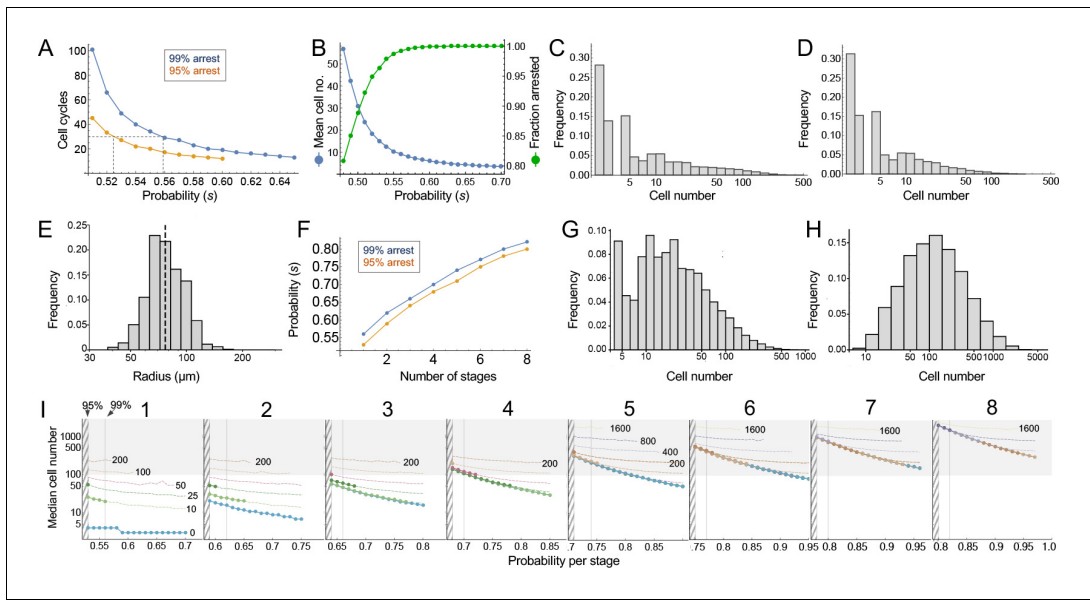

**Figure 4.** Modeling cell-autonomous clonal arrest as a probabilistic process. (A-D) Monte Carlo simulations were carried out in which a single-cell replicates and arrests with fixed probability, *s*, per cell cycle. (A) Cell cycles required before proliferation stops in 95% or 99% of simulations. (B) Mean cells at 30 cell cycles, and fraction of clones expected to have arrested by then. (C–D) Clone (nevus) size distributions, after 30 cell cycles, assuming *s* = 0.56 (C), the probability required for 99% arrest, or 0.53 (D), the probability required for 95% arrest (see panel A). (E) Actual mouse nevus sizes at P21 (mice = 3, nevi = 768). Dashed line shows median radius, 76.8 μm. (F–I) Simulations in which proliferating cells arrest after multiple events (stages). (F) The value of *s* required to ensure arrest within 30 cell cycles, as a function of number of stages. (G–H) Clone size distributions for two (G), or three (H) stages, assuming the lowest per-stage transition probability compatible with 99% arrest by 30 cell cycles (see panel C). (I) Median clone size for different numbers of stages (labeled above each graph), transition probabilities per stage (plotted on the abscissa), and thresholds below which clones are excluded from analysis. Each curve represents a different exclusion threshold (between 0 and 200 cells for 1–4 stages, and up to 1600 cells for 5–8 stages, as labeled). Curves change from solid to dashed where the observability threshold becomes inconsistent with the observations in panel E (in panel E the median nevus has >3 times the area of the smallest observable nevi; in panel I the curves become dashed when median cell number is less than twice the observability threshold). Within the hatched region, fewer than 95% of clones arrest by 30 cell cycles. The thin line to right of the hatched region marks the probability at which 99% of clones arrest by 30 cell cycles. Solid gray demarcates median cell numbers between 100 and 3000 (see text). All results are from a minimum of 20,000 simulations.

The online version of this article includes the following source data and figure supplement(s) for figure 4:

**Source data 1.** Raw data used to generate histogram in *Figure 4E*.

**Figure supplement 1.** Size distribution of mouse and human nests.

**Figure supplement 1—source data 1.** Raw data used to generate histograms in *Figure 4—figure supplement 1*.

---

a priori way to know what threshold to use, we examined the entire range of plausible truncations (up to the largest clone sizes). As it turns out, the relative shapes of simulated distributions were about the same regardless of where they were truncated. The reason for this behavior can be understood by displaying simulated distributions (with bin sizes of one cell) on a log-log plot, and observing that they fall, over most clone size ranges, on a straight line (see Supplemental Material). This implies an approximately 'power law' relationship which, by definition, is scale-free, that is has the same relative shape over any range of observations. In fact, for *s* reasonably close to 0.5, the approximate probability of observing any clone size can be shown analytically to vary inversely with the 3/2-power of size (for derivation, see Appendix 1).

These data imply that observed nevus size distributions cannot be generated by any cell-autonomous, random, time independent, one-step process. But they do not speak to whether a more complicated random process, for example, one with several steps, might suffice. To address this, one can simulate clonal evolution under multi-step models. Again, dynamic predictions can be made. First, to achieve clonal stopping times within 30 cell cycles, the minimum per-step transition

probability increases with the number of steps (*Figure 4F*). For example, if it takes three random events to arrest growth in 99% of clones, the average probability of each event needs to be at least 0.66 per cell cycle; with five random events that number is 0.74.

Second, although distributions generated by such models still tend to be heavy-tailed (*Figure 4G*), they become less so as the number of steps increases (*Figure 4H*), gradually approaching something that looks log-normal. This makes mathematical sense: as per-cell-cycle probabilities approach unity, the system approaches a clock that simply ticks off a fixed number of cell cycles before stopping. A scenario in which all clones stop at roughly the same time, plus or minus some variation, necessarily produces a log-normal distribution, since the logarithm of cell size will be proportional to the number of cell divisions.

To determine how many independent steps would be required for a random cell-autonomous process to produce distributions that fit those we observed for nevi, we simulated up to eight random stages, over a variety of transition probabilities (*Figure 4I*). We subjected the results to a range of possible truncations, from 0 to 1600 cells, to mimic any observability cutoffs in the empirical data, and recorded the median clone sizes produced under each of these scenarios.

As described below (see Materials and methods), we estimate that the average nevus has about 500–1000 cells, but given possible errors in the estimate, we consider here a range of values between 100 and 3000 (gray-shaded area in *Figure 4I*). Subject to the constraint that enough clones must arrest within 30 cell cycles, and that observation thresholds cannot be so high that the observed median is less than twice the threshold, we find that, to produce nevi of even 200 cells requires 4–5 independent events (stages), depending on whether one requires 95% or 99% clonal arrest; to reach 500 cells requires 6–7 events. To reach even larger numbers—as would be found in human nevi, or in other mouse models (*Chai et al., 2014*)—would require even more stages.

## Does a collective process arrest nevi?

The above results indicate that, to generate *in vivo*-like distributions of nevi, a process something like a clock is needed, with cells either counting elapsed divisions (or time) since oncogene activation, or progressing through a sufficiently large sequence of random processes, with tightly controlled probabilities, so that the net outcome is clock-like.

Cell-autonomous counting of cell cycles (up to about 12) can occur in early, cleavage-stage embryos (*Tadros and Lipshitz, 2009*), but no mechanism has been described to enable growing (as opposed to merely cleaving) cells to track more than a small handful of divisions (or the equivalent amount of time). Erosion of telomeres can mark the passage of large amounts of time in some cells, but this does not seem to occur to any significant degree in nevus melanocytes (*Michaloglou et al., 2005*).

In contrast, if growth arrest is not cell-autonomous, but driven by cell–cell communication, then clock-like behavior is easily achieved, without any sort of intrinsic cell memory: Consider a simple communication circuit in which every cell's arrest probability is simply a monotonically increasing function of the number of cells around it that have already arrested (*Figure 5A*). This mechanism describes a dynamically well-understood feedback process that normal tissues use to control size (*Lander, 2011*; *Lander et al., 2009*). Termed 'renewal control' (*Buzi et al., 2015*), because differentiated cells control the probability that progenitor cells self-renew, the process is often mediated by secreted TGF-β superfamily members such as myostatin, activin and GDF11 (*Gokoffski et al., 2011*; *Lee et al., 2005*). Because it implements the engineering principle of 'integral negative feedback', renewal control produces highly robust final population sizes that are independent of parameters such as cell cycle speed or the starting numbers of cells (*Buzi et al., 2015*; *Lander, 2011*; *Lander et al., 2009*).

When growth arrest due to renewal control is simulated as a probabilistic process (*Figure 5B*), the observed size distributions of clones are very close to log-normal. This is because renewal control effectively enforces cell cooperation, so that once a small fraction of a clone has arrested, the entire clone stops soon thereafter. The resulting narrow distribution of stopping times produces size distributions that are approximately log-normal, that is that emulate a clock.

This behavior is a generic outcome of feedback control, and does not depend strongly on the details of how feedback is implemented. Similar distributions are obtained whether we model nevi as progressing reversibly or irreversibly through more than one proliferative stage, or use agent-based simulations in which renewal is controlled by the concentration of a secreted molecule that

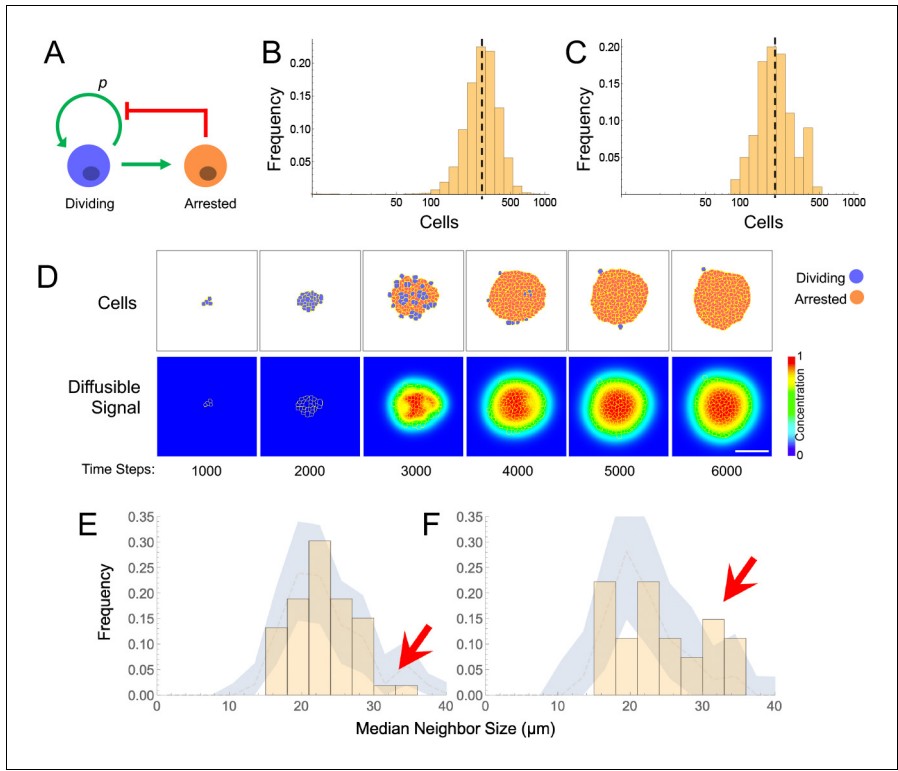

**Figure 5.** Models and evidence for cooperative, feedback-mediated arrest. (**A**) A generic integral negative feedback scheme. 'Renewal probability', $p$, is the probability that offspring of cell division remain dividing (i.e. $1-p$ is the probability that they arrest). (**B**) Clone sizes generated by 9115 stochastic simulations of scheme A, modeled as an ordinary differential equation, where $p$ falls with the number of arrested cells according to a Hill function with half-maximal effect at 50 arrested cells. (**C–D**) Results from a spatial (agent-based) simulation of scheme A, in which the signal from arrested cells spreads by diffusion. Histogram (**C**) tabulates clone sizes produced by 100 independent simulations (the histograms in both B and C are logarithmically-scaled to show that the data are well fit by log-normal distributions). Panels (**D**) are from a single simulation, showing locations of growing and arresting cells, and the gradient of the diffusible signal. Bar = 50 µm. The average cell cycle is equivalent to approximately 382 Monte Carlo time steps. (**E–F**) Using spatial coordinates and areas of 122 nests in seven individual fields at P21, nests were modeled as disks of equivalent area, and mean sizes of neighboring nests falling within successively larger annuli around each target nest were determined. Distributions of mean neighbor sizes up to 45 µm away from (**E**) large (radius >20 µm) and (**F**) small (radius <20 µm) nests (histograms) are compared with a 'null distribution' derived by random permutation (blue fields). Arrows show deviations in neighbor size distribution greater than expected at random, and substantially different for large versus small nests.

The online version of this article includes the following source data and figure supplement(s) for figure 5:

**Source data 1.** Raw data used to generate histograms and permutation tests in *Figure 5E–F*.
**Figure supplement 1.** Evidence for influence of nevi on each other's sizes is minimal at long range.

accumulates according to the laws of diffusion and local uptake (*Figure 5C–D*). The point of these simulations is not to argue in favor of a specific feedback mechanism, but rather to show that, where cell-autonomous mechanisms of arrest struggle to fit nevus dynamics, almost any sort of (collective) feedback does so easily.

Although nevus size distributions alone cannot shed light on the molecular details of how feedback might be implemented in nevi, it is interesting that those cells that we identify as nevus melanocytes (*Figure 2*) express multiple genes encoding ligands with known or suspected growth inhibitory activities, together with the receptors for those ligands. These include TGFβ superfamily members *Gdf11*, *Gdf15*, *Tgfb1*, *Tgfb2*, and *Tgfb3*, as well as other genes associated with growth inhibition, such as *Angptl2*, *Angptl4*, *Il6*, *Sema3a*, *Sema3b*, and *Sema3f* (*Attisano and Wrana, 1996*; *Neufeld et al., 2016*; *Santulli, 2014*).

The evaluation of such candidates (as well as other genes expressed at too low a level to be reliably detected by single-cell RNA sequencing) will no doubt require further study. In the meantime, we reasoned that any feedback mechanism based on secreted, diffusible factors should induce spatial correlations among clones. In particular, when clones (or subclones) get close enough to each other, they should inhibit each other's growth, leading to a smaller final size. The distance over which such effects could occur should reflect the spatial decay length of diffusible molecules in the skin, which is thought to be on the order of no more than a few hundred microns (*Chen et al., 2015*). Although our data on macroscopic nevi (*Figure 1D*), which had been collected in a manner that included spatial coordinates, did not contain enough examples of nevus spacings in this range to test this hypothesis, our data on the nests within individual nevi did, as the median spacing between nests at postnatal day 21 was approximately 79 microns.

To assess whether nests were significantly smaller when located near other nests (especially large ones), we extracted the coordinates and nest areas from seven separate fields at P21 (representing 122 individual nests) and, modeling each nest as a disk of equivalent area, calculated the mean sizes of neighboring nests falling within successively larger annuli around each target nest. We compared the distributions of mean neighbor sizes near the 52 smallest nests (radius <20 μm) with the equivalent distributions around the 70 largest nests.

As shown in *Figure 5E–F*, within annuli extending 45 μm away from the perimeters of target nests, we saw fewer examples of large neighbors (radius >30 μm) around large nests (panel E) than around small ones (panel F). To determine whether the difference was statistically significant, we used a permutation test in which we randomly swapped nest areas (but not locations) within each field 5000 times, and recalculated the distributions. This allowed us to plot the envelope enclosing the 5th and 95th percentiles for permuted data, onto which we overlaid the observed data. Unlike the observed data, the envelopes of the permuted data (blue zones in *Figure 5E–F*) look similar whether target nests are large or small. Moreover, the observed data extended outside of the envelopes only for median neighbor sizes > 30 μm, with the data for small target nevi extending well above the relevant envelope and the data for large target nevi lying at to the bottom of the envelope (*Figure 5E–F*, arrows). These results argue that proximity is associated with a small, but significant decrease in nest size, supporting the view that nests inhibit each other's growth. Interestingly, when we repeated the same analysis using annulus sizes of 150 μm, differences in the sizes of neighbors of small and large nests were not seen, consistent with the view that whatever is promoting coordination among nests has a spatial range of <150 μm (*Figure 5—figure supplement 1*).

## Discussion

Studies in man, mouse and fish establish that most melanocytic nevi form by mutational activation of BRAF, which triggers proliferation followed by growth arrest (*Damsky and Bosenberg, 2017*; *Dankort et al., 2009*; *Dhomen et al., 2009*; *Kaufman et al., 2016*; *Michaloglou et al., 2005*; *Patton et al., 2005*; *Shain and Bastian, 2016*). Nevus growth is often considered a paradigmatic example of OIS, but here we question two of the major tenets of the OIS hypothesis: that nevus melanocytes are actually senescent; and that growth arrest is a direct effect of oncogene action on the individual cell.

To assess whether nevi are senescent, we used single-cell RNA sequencing in a mouse model of *Braf*-driven nevus formation, comparing gene expression of nevus melanocytes with that of other cell types. Across a wide variety of gene expression signatures, especially those developed to distinguish senescence from other growth-arrested cell states, we failed to find any evidence in support of the OIS hypothesis. By gene expression criteria, nevus melanocytes were less senescent than many other normal skin cells, including non-nevus melanocytes (*Figure 3*, and its supplements). These results support earlier work that also questioned, based on immunohistochemical staining of human nevi for markers including lysosomal β-galactosidase, Ki67, p16INK4a (*CDKN2A*), γ-H2AX and p53, whether nevus melanocytes should be considered senescent (*Tran et al., 2012*). In agreement with other studies, we do find that *Cdkn2A* is highly expressed in nevus cells; it is in fact the only 'classical' senescence marker that clearly distinguishes nevus melanocytes from other melanocytes (*Figure 3A*). Yet, as others have shown, *Cdkn2A* is neither necessary nor sufficient for oncogene-mediated melanocyte growth arrest (*Haferkamp et al., 2009*; *Zeng et al., 2018*). Thus, to the

extent that nevus melanocytes do execute even part of a common senescence program, there is little to support the view that this why they stop proliferating.

As for the question of exactly how *Braf*-induced nevus growth arrest occurs, *Figure 6* presents a continuum of models: In model A, oncogene action elicits a cell-autonomous stress response which, after some time lag, triggers senescence that shuts proliferation down. This is the form in which the OIS hypothesis is most frequently presented.

In model C, growth arrest is not a direct effect of oncogene action, but rather a consequence of growth itself. This type of feedback is commonly used by adult tissues to maintain constant size, and also enables developing tissues to produce precise numbers of differentiated cells (*Kunche et al., 2016*; *Lander, 2011*; *Lander et al., 2009*). Because of the collective nature of this mechanism—cells that have stopped dividing tell other cells in their vicinity to do likewise—it naturally produces semi-synchronous arrest of spatially-coherent cell clones, and the distinctive log-normal clone size distribution that comes along with that. In contrast, a purely cell-autonomous mechanism (panel A) has great difficulty producing such distributions (*Figure 4*), either necessitating the operation of some kind of multi-cell-cycle clock, or requiring cells to complete a long sequence of independent probabilistic events prior to arresting (*Figure 4*).

One can, of course, build a model in between these two extremes (model B), in which oncogenes induce growth arrest directly, but paracrine signals (i.e. SASP factors) help maintain it. If the paracrine role is important enough, this mechanism might also produce clone size distributions that are approximately log-normal, so we cannot categorically rule this model out. However, our gene expression data do not support any versions of it that have been explicitly proposed for nevi. So far, several groups (working predominantly from in vitro observations) have claimed a critical role for SASP factors in melanocyte OIS: *Wajapeyee et al., 2008* argued that IGFBP7 plays a necessary role in the establishment of BRAF-V600E-induced melanocyte senescence (a conclusion disputed by some [*Scurr et al., 2010*]); *Feuerer et al., 2019* proposed that MIA (melanoma inhibitory activity) secreted by senescent melanocytes is required to maintain senescence; and Damsky and Bosenberg have proposed that IL1, IL6, IL8 (encoded in mouse by *Cxcl15*), and type 1 interferons produced by nevus cells play a role in their arrest (*Damsky and Bosenberg, 2017*).

Our *in vivo* data do not support any of these hypotheses. For example, we observed that the vast majority of *Igfbp7* transcripts are produced by fibroblasts and endothelial cells and that, among melanocytes, nevus melanocytes express lower levels of *Igfbp7* than non-nevus melanocytes. We did not detect any *Mia* transcripts in nevus melanocytes, although it was expressed at detectable levels

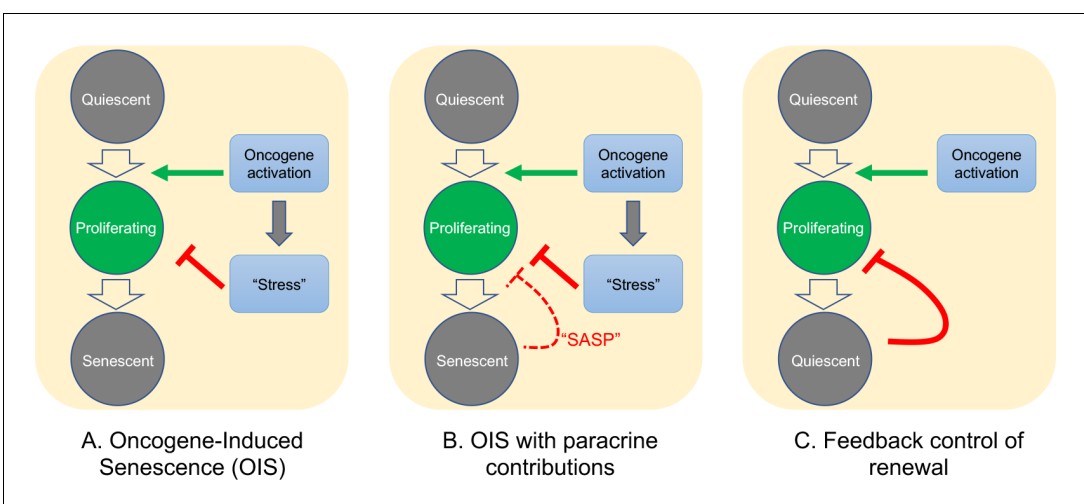

**Figure 6.** Possible mechanisms of growth arrest. Three different models of nevus growth arrest are presented, varying from classical OIS (**A**) to feedback control of proliferative cell renewal (**C**). The model in the middle panel (**B**) illustrates a hypothetical hybrid situation, in which paracrine effects of senescence-associated secreted proteins (SASP) act as inhibitors of melanocyte self-renewal. Although model B can mimic some of the dynamic behaviors of model C, in the absence of convincing evidence that nevus melanocytes actually are senescent, we favor model C.

in non-nevus melanocytes and various other skin cells. Likewise, of *Il1* family members, only *Il1a* transcripts were detected in nevus melanocytes and they were at levels lower than in many other skin cell types. *Il6* was also only weakly expressed in nevus melanocytes, especially when compared with other cells. Transcripts for type one interferons were not detected in any melanocytic cells, and *Cxcl15* transcripts were not detected in any skin cells at all.

Of course, the accuracy of single-cell RNA sequencing can be limited for weakly expressed genes, so we cannot completely eliminate the possibility that these factors play some role in nevus growth arrest. But given these results, and the evidence that nevus melanocytes are not senescent, we strongly favor the renewal-feedback model (model C). Adopting this model also makes it easier to accommodate long-standing evidence that nevus growth arrest is not permanent (*Shain and Bastian, 2016*). For example, it is known that nevi may exhibit a low level of mitoses (*Glatz et al., 2010*); that they can grow in response to stimuli such as UV light (*Rudolph et al., 1998*) or immunosuppression (*Shain and Bastian, 2016*) and, perhaps most tellingly, they can re-grow after incomplete surgical resection—stopping again when they reach a typical nevus size (*Vilain et al., 2016*). The latter result is inherently problematic for any non-feedback model, but is precisely what renewal feedback predicts (*Lander, 2011*; *Lander et al., 2009*).

Because feedback control of renewal implements a generic strategy (integral negative feedback [*Lander, 2011*]), it places no constraints on the molecular details of feedback, short of the fact that whatever is mediating it needs to rise with the number of cells already arrested. One possibility is that nevi are responding to some of the same signals that are used in melanocyte homeostasis. For instance, during anagen phase of the hair cycle, melanocyte stem cells produce progeny that migrate out of the hair follicle bulge as they differentiate, leaving functional stem cells behind for future cycles. A variety of experimental and pathological circumstances that allow small numbers of melanocytes to differentiate within the bulge cause differentiation and loss of the entire stem cell pool (with concomitant hair graying [*Nishimura et al., 2005*]). This sort of behavior—where differentiated cells drive the differentiation of their progenitors—is exactly the sort of behavior that drives feedback models of renewal (*Buzi et al., 2015*; *Lander, 2011*; *Lander et al., 2009*).

Nevi are but one of many types of benign, clonal, proliferative lesions that arise due to the activation of oncogenes, but rarely if ever progress to malignancy (*Adashek et al., 2020*). Notwithstanding the disruptive influence that oncogenes can have on cell physiology, the existence of such lesions suggest that homeostatic mechanisms persist and function at many stages along the road to cancer. New avenues for cancer prevention and treatment are likely to follow from the detailed elucidation of such mechanisms.

# Materials and methods

## Key resources table

| Reagent type (species) or resource | Designation | Source or reference | Identifiers | Additional information |
|---|---|---|---|---|
| Gene (*Mus musculus*) | Braf | Mouse Genome Informatics (MGI) | MGI:88190 | |
| Gene (*M. musculus*) | Tyr::CreER | MGI | MGI:3641203 | MGI Transgene name: GeneTg(Tyr-cre/ERT2)13Bos |
| Genetic reagent (*M. musculus*) | B6.Cg-Tg(Tyr-cre/ERT2) 13Bos Braf^tm1Mmcm /BosJ | *Dankort et al., 2009* | RRID:MGI:5902125 | |
| Antibody | Anti-Pmel (rabbit monoclonal) | Abcam | Cat#ab137078 RRID:AB_2732921 | Also known as anti-melanoma gp100 IF (1:500) |
| Antibody | Anti-BrdU (rat monoclonal) | Abcam | Cat#ab6326 RRID:AB_305426 | IF (1:500) |
| Antibody | Goat anti-rabbit alexa fluor 594 conjugated (polyclonal) | ThermoFisher Scientific | Cat#A-11012 RRID:AB_2534079 | (1:2000) |
| Antibody | Chicken anti-rat alexa fluor 488 (polyclonal) | ThermoFisher Scientific | Cat#A-21470 RRIB:AB_2535873 | (1:2000) |

*Continued on next page*

*Continued*

| Reagent type (species) or resource | Designation | Source or reference | Identifiers | Additional information |
|---|---|---|---|---|
| Sequence-based reagent | Braf_F | IDT | PCR primer | 5'-TGAGTATTTTTGTGGCAACTGC −3' |
| Sequence-based reagent | Braf_R | IDT | PCR primer | 5'-CTCTGCTGGGAAAGCGCC −3' |
| Sequence-based reagent | Cre_F | IDT | PCR primer | 5'- GGTGTCCAATTTACTGACCGTACA-3' |
| Sequence-based reagent | Cre_R | IDT | PCR primer | 5'- CGGATCCGCCGCATAACCAGTG −3' |
| Chemical compound, drug | 4-hydroxytamoxifen | Sigma-Aldrich | Cat#68047-06-3 | |
| Chemical compound, drug | TrueBlack lipofuscin | Biotium | Cat#23007 | |
| Software, algorithm | Mathematica | Wolfram | RRID:SCR_014448 | |
| Software, algorithm | Scanpy | *Wolf et al., 2018* | RRID:SCR_018139 | |
| Software, algorithm | Cell Ranger | 10X genomics | RRID:SCR_017344 | |
| Software, algorithm | CompuCell3D | *Swat et al., 2012* | RRID:SCR_003052 | |

## Mouse treatment for nevus development

*Braf*$^{V600E}$, *Tyr-Cre*ER (C56BL/6) mice (RRID:MGI:5902125) were genotyped by PCR as previously described (*Bosenberg et al., 2006*; *Dankort et al., 2007*). The primers used in this study are: Braf forward 5'-TGAGTATTTTTGTGGCAACTGC −3', Braf reverse 5'-CTCTGCTGGGAAAGCGCC −3', Cre forward 5'- GGTGTCCAATTTACTGACCGTACA-3' and Cre reverse 5'- CGGATCCGCCGCATAACCAGTG −3'. Topical administration of 4-hydroxytamoxifen (4-OHT; 25 mg/mL or 75 mg/mL in DMSO; 98% Z-isomer, Sigma-Aldrich) was administered to pups on their back and/or paws at ages P2, P3, and P4. Images of nevi on back and paw skin were taken with a digital camera at the indicated ages. Nevi from the underside of the skin were imaged using a dissection microscope. All mouse procedures were approved by UCI's IACUC.

## Live imaging of the skin by MPM

Mice were sedated, shaved, and depilated with wax strips at the indicated ages (during a telogen phase) and the dorsal skin was imaged to capture the intrinsic fluorescent signal from keratin, melanin, as well as the second-harmonic-generation signal from collagen, using the LSM 510 NLO Zeiss system. Excitation was achieved with a femtosecond Titanium: Sapphire (Chameleon-Ultra, Coherent) laser at 900 nm. Emission was detected at 390–465 nm for second harmonic generation (blue) and 500–550 nm (green) and 565–650 (red) for fluorescence.

## *In vivo* labeling with BrdU

BrdU was prepared in sterile PBS at 10 mg/mL and injected intraperitoneally into mice that were 20 days old at 100 mg/kg of body weight. 24 hr later the mouse was shaved, depilated with wax strips and the skin was removed and fixed in 10% formalin for 16 hr.

## Immunofluorescence

Formalin fixed paraffin embedded skins were sectioned 8 μm thick, deparaffinized with Xylene, and dehydrated in a series of increasing concentration of ethanol washes. Antigen retrieval was performed with 10 mM citric acid buffer at pH 6.0 for 10 min in a steamer. Samples were washed with PBS, incubated with TrueBlack for 30 s to reduce autofluorescence, and washed again with PBS. All antibodies were diluted at a 1:500 and incubated overnight at 4°C. Samples were washed and incubated with the appropriate secondary antibody. Melanocytes were identified with a Pmel antibody

(EP4863(2); ab137078, Abcam; RRID:AB_2732921). Cells that incorporated BrdU were visualized with a BrdU antibody (ab6326, Abcam; RRID:AB_305426).

## Cell isolation for single-cell RNA sequencing

*Braf^WT^*, *Tyr-Cre*ER or *Braf^V600E^*, *Tyr-Cre*ER mice were euthanized at either P30 (n = 2 of each genotype) or P50 (n = 3 of each genotype), shaved, and depilated. A 2 × 3 cm section of the dorsal skin was removed, and the fat scraped off from the underside. The piece was then diced into smaller pieces and suspended in dissociation buffer (RPMI, liberase 0.25 mg/mL, Hepes 23.2 mM, Sodium Pyruvate 2.32 mM, Collagenase:Dispase 1 mg/mL) for 50 min at 37°C with gentle agitation. After incubation, DNaseI (232U) was added for 10 min and then inactivated with fetal bovine serum and EDTA (1 mM). The tissue suspension was further dissociated mechanically with the GentleMACS using the setting m_imptumor_04.01, which runs for 37 s at various speeds. Single-cell suspensions were filtered twice through a 70 µm strainer and dead cells removed by centrifugation at 300 x *g* for 15 min. The live cells were washed with 0.04% UltraPure BSA:PBS buffer, gently re-suspended in the same buffer, and counted using trypan blue.

## Library preparation for single-cell RNA sequencing and analysis

Libraries were prepared using the Chromium Single Cell 3' v2 protocol (10X Genomics). Briefly, individual cells and gel beads were encapsulated in oil droplets where cells were lysed and mRNA was reverse transcribed to 10X barcoded cDNA. Adapters were ligated to the cDNA followed by the addition of the sample index. Prepared libraries were sequenced using paired end 100 cycles chemistry for the Illumina HiSeq 4000. FASTQ files were generated from Illumina's binary base call raw output with Cell Ranger's (v2.1.0; RRID:SCR_017344) 'cellranger mkfastq' command and the count matrix for each sample was produced with 'cellranger count'. All ten samples (four samples from P30 [two control (wild type) and two mutant] and six samples from P50 [three control and three mutant]) were aggregated together with the 'cellranger aggr' command to produce one count matrix that includes all samples. Data analysis was performed with Scanpy [v1.3.6; RRID:SCR_018139] (*Wolf et al., 2018*). Cells with fewer than 200 detected genes, and genes detected in less than three cells, were discarded. We calculated the percent mitochondrial gene expression and kept cells with less than 13% mitochondrial gene expression, and cells with fewer than 4000 genes/cell (35,141 cells). Each cell was normalized to total counts over all genes. In the final preprocessing step, we regressed out cell-cell variation driven by mitochondrial gene expression and the number of detected UMI. To identify clusters, we first performed principal component analysis on log-transformed data, using highly variable genes, Louvain clustering (*Levine et al., 2015*), and visualization with t-distributed stochastic neighbor embedding (tSNE).

## Quantification of nevus and nest size and cell content

To quantify the sizes of nevi in mice, dorsal skin was excised and the underside visualized using a dissecting microscope. Nevi were traced, and area calculated using ImageJ. Nest sizes were quantified in live mice by MPM. Sizes of human nests were measured from histological samples (n = 5) obtained from the UCI Department of Dermatology. Samples were stained with hematoxylin and eosin and imaged with a microscope. A dermatologist manually identified the nests on each slide, and nest area was quantified using ImageJ. Human studies were performed under IRB protocol HS# 2019–5054.

Estimates of cell numbers for mouse nevi were obtained in two different ways: First, we used estimates from *Chai et al., 2014* for melanocytic nuclei per square area of mouse nevus, together with our observed median nevus radius of 76.8 µm; this approach led to an estimate of 897 cells/nevus. As the data of *Chai et al., 2014* come from a different genetic model, we also estimated cell number as follows: Using 8 µm sections of back skin from Albino Braf^V600E^ mice, we used fluorescence microscopy to measure the sizes of 194 Pmel-stained melanocytes within the nests of nevi, obtaining an average cell diameter of 5.68 µm, and counted approximately 14.4 cells per $10^4$ µm$^3$ of nest. In pigmented animals, we measured by MPM an average nest cross sectional area of 1385 µm$^2$, an average nest volume of 38792 µm$^3$, and an average number of nests per nevi of approximately 12, yielding an estimate of 672 cells/nevus. Given uncertainties in these measures, analyses in the manuscript take into account the possibility of an average that falls anywhere between 100 and 1000 cells.

## Simulations and agent-based modeling

Stochastic, non-spatial simulations of renewal control were obtained by Monte Carlo simulation, in which cells duplicated every cell cycle, and then chose randomly whether to differentiate or continue dividing according to a probability modified by feedback from non-dividing cells. A Hill function, with Hill coefficient = 1, was used to represent the feedback.

To model feedback in a spatial context, we used CompuCell3D (RRID:SCR_003052), an open-source platform for Cellular Potts modeling (*Swat et al., 2012*). In CPM, every generalized object or 'cell' is associated with a list of attributes such as cell type, surface area, volume, etc. These enter into the calculation of an effective energy, which can be summarized as the sum of the contact energy between neighboring cells and the effective energy due to volume constraints.

Simulations were initialized by seeding a single cell, with a size of 25 pixel$^2$, in the center of a 300 $\times$ 300 pixel lattice, which grew and divided according to rules and parameters summarized in *Source data 2*. To add variability to cell growth, cells randomly chose one of three different growth rates after every cell division. To add variability to cell division times, cells randomly chose a target area, between 72 and 80 pixel$^2$, at which to divide. Growth rates were chosen to be sufficiently slow that the mean time between cell divisions came out to approximately 382 time steps. At division, each cell was divided in half by a randomly-oriented division plane.

Upon division, a cell either remained dividing or became permanently arrested. All cells had a minimum 1% probability of arrest per cell division. Once an arrested cell was generated, it began continuous secretion of a signaling molecule that diffuses and promotes the transition from dividing to non-dividing (*Kunche et al., 2016*). Diffusion and decay of the feedback factor was modeled deterministically, with parameters chosen to produce a steady state decay length of 15 pixels. The concentration of this factor at the center of mass of each cell then augmented the arrest probability of that cell by an amount determined by a Hill function (see *Source data 2*).

## Statistical analysis

Statistical analyses for single-cell RNA sequencing were performed using Scanpy (RRID:SCR_018139). Other statistical testing was done using *Mathematica* (RRID:SCR_014448). For the spatial analysis in *Figure 5E–F*, nest areas in each field were randomly swapped, with positions held constant, 5000 times, and the distributions of neighboring nest locations and sizes recalculated each time. This allowed us to generate an envelope enclosing the 5[th] and 95[th] percentiles for the permuted data, at each target nest size, and compare the observed data with the bounds of that envelope.

## Acknowledgements

The authors are grateful to Eric Mjolsness, Allon Klein, Randy Heiland and Paul Macklin for helpful discussions. **Funding:** This work was supported by National Institute of Health (NIH) grants U54CA217378 and P30AR075047 (ADL, AKG, JL). MGC was supported by NiH-T32EB009418. RRV was supported by the UC Presidents fellowship, FORD Foundation Fellowship and NIH-T32CA009054. **Author Declaration of interests:** No conflict of interest declared by any author. **Data Availability:** Raw sequence data are available at GEO (GSE154679).

## Additional information

### Funding

| Funder | Grant reference number | Author |
|---|---|---|
| National Cancer Institute | U54CA217378 | John Lowengrub<br>Anand K Ganesan<br>Arthur D Lander |
| National Institute of Arthritis and Musculoskeletal and Skin Diseases | P30AR075047 | Arthur D Lander<br>Anand K Ganesan<br>John Lowengrub |
| National Institute of Biomedical Imaging and Bioengineer- | T32EB009418 | Michael G Caldwell |

ing

| University of California | Postdoc Fellowship | Rolando Ruiz-Vega |
| National Academies of Sciences, Engineering, and Medicine | Postdoc Fellowship | Rolando Ruiz-Vega |
| National Cancer Institute | T32CA009054 | Rolando Ruiz-Vega |

The funders had no role in study design, data collection and interpretation, or the decision to submit the work for publication.

### Author contributions
Rolando Ruiz-Vega, Conceptualization, Data curation, Formal analysis, Investigation, Writing - original draft, Writing - review and editing; Chi-Fen Chen, Data curation, Formal analysis, Investigation; Emaad Razzak, Priya Vasudeva, Tatiana B Krasieva, Jessica Shiu, Data curation, Investigation; Michael G Caldwell, Formal analysis, Investigation; Huaming Yan, Formal analysis; John Lowengrub, Methodology; Anand K Ganesan, Conceptualization, Supervision, Funding acquisition, Writing - original draft, Writing - review and editing; Arthur D Lander, Conceptualization, Formal analysis, Supervision, Funding acquisition, Methodology, Writing - original draft, Project administration, Writing - review and editing

### Author ORCIDs
Anand K Ganesan (ID) https://orcid.org/0000-0003-4944-9274
Arthur D Lander (ID) https://orcid.org/0000-0002-4380-5525

### Ethics
Animal experimentation: This study performed in strict accordance with the recommendation from University Laboratory Animal Resources (ULAR). All the animals were handled according to approved institutional animal care and use committee (IACUC) protocol (#AUP-17-230) at the University of California Irvine.

### Decision letter and Author response
Decision letter https://doi.org/10.7554/eLife.61026.sa1
Author response https://doi.org/10.7554/eLife.61026.sa2

## Additional files

### Supplementary files
- Source data 1. Gene lists and literature references for gene expression signatures.
- Source data 2. Parameters values used in CompuCell3D modeling.
- Transparent reporting form

### Data availability
Single cell RNA sequencing data have been deposited in GEO under accession number GSE154679. Parameters for simulations are found in Data S2 in the manuscript.

The following dataset was generated:

| Author(s) | Year | Dataset title | Dataset URL | Database and Identifier |
|---|---|---|---|---|
| Ruiz-Vega R, Ganesan AK, Lander AD | 2020 | Single cell gene expression of melanocyte specific Braf mutant mouse skin | https://www.ncbi.nlm.nih.gov/geo/query/acc.cgi?acc=GSE154679 | NCBI Gene Expression Omnibus, GSE154679 |

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

## Appendix 1

### Modeling Oncogene-Induced Senescence as a Cell Autonomous Process—Mathematical Results

Consider the clonal descendants of an oncogene-transformed cell. The simplest model of a cell-autonomous oncogene-induced arrest process is one in which cells replicate at a constant rate, and undergo senescence with a constant probability $s$ per cell cycle. Once all the cells in a clone have undergone senescence, we refer to the clone as 'extinct'. For any given $s$, we wish to find the probability that extinction has occurred by any given time, as well as the distribution of clone sizes that should be expected at that time.

If the decision to senesce is independent in each daughter cell at each cell division, then this scenario describes a branching process in which each cell produces two senescent cells with probability $s^2$, two dividing cells with probability $(1-s)^2$, and one dividing and one senescent cell with probability $2s(1-s)$. The theory of branching processes may then be used to obtain the probability generating function (PGF) for the number of dividing cells after $n$ cell cycles. The offspring distribution of a single cell as described above may be written as:

$$f(z) = s^2 + 2s(1-s)z + (1-s)^2 z^2$$

where $z$ is a dummy variable whose exponent indicates the number of dividing (non-senescent) cells produced by an event, and the coefficient in front of each $z^n$ is the probability of that event. It can be shown that the PGF for the number of dividing cells after $n$ cell cycles is equal to the $n$-th composition of the offspring distribution $f$ unto itself (*Hawkins and Ulam, 1944*). For example, the PGF for the number of dividing cells after 2 cell cycles is:

$$F(2, z) = f(f(z)) = s^2 + 2s(1-s)(s^2 + 2s(1-s)z + (1-s^2)z^2) \\ + (1-s)^2(s^2 + 2s(1-s)z + (1-s)^2 z^2)^2$$

And in general:

$$F(n, z) = F(1, F(n-1, z)) = f(F(n-1, z))$$

Thus $F(n, 0)$ is the cumulative probability that there are 0 dividing cells—that is a clone has extinguished—after $n$ cell cycles.

The theory of branching processes also tells us that the probability that a clone eventually goes extinct (over the long run) is just the smallest non-negative root of $z = f(z)$. We solve the equation: $z = s^2 + 2 s(1-s)z + (1-s)^2 z^2$ and obtain: $z = s^2/(1-s)^2$, $0 \leq s < 0.5$ and $z = 1$, $s \geq 0.5$. This confirms that only if $s \geq 0.5$ is eventual extinction of all clones guaranteed.

### Monte Carlo Simulations

The behavior of the above branching processes may be easily observed using Monte Carlo Simulation, seeding each clone with a single dividing cell and specifying the value of $s$. In the simulations show in *Figure 4*, cells divided synchronously and $s$ was used to determine the fate of each daughter cell after each division. For each simulation it was recorded (1) whether the clone went extinct, and if so, after how many cell cycles it did so and (2) the number of cells at the time of extinction (or the end of the simulation).

### Distribution of clone sizes at extinction

To derive the distribution of clone sizes at extinction it is helpful to think about clonal development not in terms of the number $n$ of cell cycle times that have elapsed, but in terms of the cumulative number of cell division events that have occurred up to any given time within a clone, which we will represent as $\eta$. Assuming clones begin from one cell, the number of cells at extinction ($T_c$) will simply be $T_c = 1 + \eta$;

To find the expected distribution of values of $T_c$, we begin by computing the probability that a clone goes extinct at a given value of $\eta$, which we will call $p_E(\eta)$. Recalling that two choices are made at every division (one per daughter cell), it may be seen that to extinguish at $\eta$ requires $2\eta$ choices,

exactly $\eta + 1$ of which are choices to become senescent, and $\eta - 1$ events are choices to remain proliferative. This implies that:

$$
\begin{aligned}
p_E(1) &= A_1 s^2 \\
p_E(2) &= A_2 (1-s) s^3 \\
p_E(3) &= A_3 (1-s)^2 s^4 \\
&\cdots \\
p_E(\eta) &= A_\eta (1-s)^{\eta-1} s^{\eta+1}
\end{aligned}
$$

where the coefficients $A_\eta$ are constants that capture the number of different ways that each combination of choices of $s$ and $1-s$ can happen. $A_\eta$ can be thought of as the number of unique full binary trees that end with $\eta + 1$ senescent cells. As senescent cells are the dead ends in those trees, they are referred to as 'leaves' of the tree. The sequence that counts the number of unique full binary trees is called the *Catalan numbers*. $C_k$, the $k$th Catalan number is the number of unique binary trees with $k+1$ leaves. It is defined by:

$$
C_k = \frac{1}{k+1} \binom{2k}{k}
$$

Accordingly,

$$
p_E(\eta) = \frac{1}{\eta+1} \binom{2\eta}{\eta} (1-s)^{\eta-1} s^{\eta+1}
$$

Consequently, the probability of $T_c = m$ cells at extinction is simply

$$
p_{T_c}(m) = p_E(m-1)
$$

In *Appendix 1—figure 1*, panel A, we plot the value of $p_E(\eta)$ as a function of $\eta$ for different values of $s$, using logarithmic axes. Notice that for $s$ sufficiently close to 0.5, the relationship approximates a line of slope $-3/2$. Thus, the approximate probability of finding a clone of size $m$ varies inversely with the 3/2-power of $m$. In panel B, the analytical results for $s = 0.56$ are superimposed on results obtained by Monte Carlo simulation of 500,000 cases.

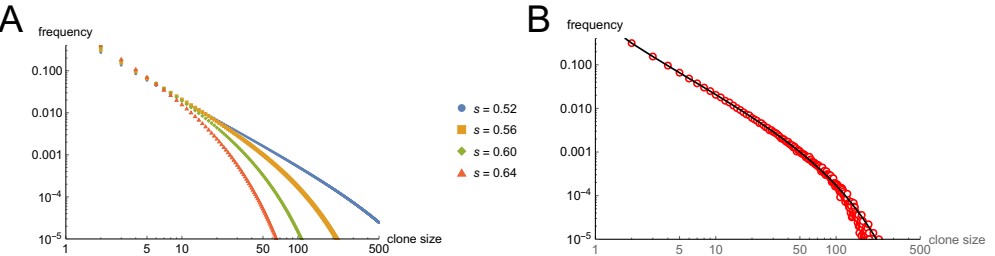

**Appendix 1—figure 1.** Predicting clone sizes at the time of clonal extinction. (**A**) Analytical results, under assumptions of different values for the cell-autonomous arrest probability $s$. (**B**). A comparison of the analytical results for $s = 0.56$ with the outcomes of 500,000 Monte Carlo simulations.

