## [Decision Letter]

**Acceptance summary:**

In this elegant manuscript, Ruiz-Vega and collaborators test the generally accepted hypothesis that nevi are growth-arrested due to oncogene-induced senescence, which is a cell-autonomous process thought to be triggered by the oncogenic mutation BRAF V600E, present in the majority of nevi. They analyse single-cell transcriptome data from mice nevi and do not find any evidence of these cells being senescent. Rather, and showing great creativity, they find through the use of mathematical models that the distribution of nevi sizes is better explained by cooperative mechanisms, of the kind used in size control of normal tissues.

**Decision letter after peer review:**

Thank you for submitting your article "Dynamics of nevus development implicate cell cooperation in the growth arrest of transformed melanocytes" for consideration by *eLife*. Your article has been reviewed by three peer reviewers, and the evaluation has been overseen by a Reviewing Editor and Richard White as the Senior Editor. The following individuals involved in review of your submission have agreed to reveal their identity: Peter Adams (Reviewer #1); Mariana Gómez-Schiavon (Reviewer #2); Heinz Arnheiter (Reviewer #3).

The reviewers have discussed the reviews with one another and the Reviewing Editor has drafted this decision to help you prepare a revised submission.

In this manuscript, the authors present evidence that benign nevi do not simply reflect oncogene-induced senescence. The authors propose, based on single cell RNA-seq of BRAF oncogene-expressing mouse melanocytes from mouse skin, that nevus growth arrest is more likely related to the cell interactions that mediate size control in normal tissues, than to any cell-autonomous, "oncogene-induced" program of senescence. Then, the authors propose a basic integral feedback control as the underlying mechanism behind nevi growth regulation. The editors and reviewers consider that this manuscript is exquisitely done, clear and convincing, and an important contribution to an important topic. As the authors note, the prevailing dogma is that nevus melanocytes are senescent.

All issues raised by the reviewers can be considered minor, and are listed below.

1) About the concept behind the paper:

The reviewers highlight two conceptual questions that would be in need of an answer:

a) what are the suggested mechanisms leading to growth arrest of the first cells in a clone? If this were due to some probabilistic event that, however, still depended on oncogene activation, would that not favor the hybrid model B (at least for these initial cells) rather than the authors' preferred model C?

b) While the model is clearly exclusively or nearly exclusively cell non-autonomous, it is still nevus-autonomous. The authors argue that the OIS model is mostly based on in vitro observations, but should a nevus-autonomous model not also be reproducible in vitro? If it were not, then one would have to invoke an influence of the nevus microenvironment, which would leave room for, among other factors, a senescence-associated program generated by the surrounding cells.

What would be the authors' opinion on these two points?

2) Regarding the single-cell RNAseq data:

a) The authors show data to indicate that BRAF-expressing nevi are no-proliferating, compared to WT hair follicle melanocytes. What about BRAF-expressing melanocytes in hair follicles? Does the system permit an analysis?

b) The authors compare their single cell RNA-seq data to many different senescence datasets and signatures – but most of these are from fibroblasts, including Hernandez-Segura et al., 2017. However, Pawlikowski et al., 2013, reported gene expression profiling specifically in primary human melanocytes expressing activated BRAF. Can a comparison to the dataset in this paper be made, as it is judged to be better suited?

3) Regarding the mathematical model:

a) When the authors are exploring the hypothesis of the multiple stages for a cell-autonomous growth arrest (on Figure 4 legend, and in the Results), the authors use "observation thresholds cannot be so high that the observed median is less than twice the threshold" as an argument to reject some models with few stages. Can you please explain what the logic is behind this restriction? What are the implications?

b) On Figure 4C-D, a reviewer found it difficult to compare the two models. Can you please explain what the intended message is? It might be clearer if the two distributions are overlapped.

c) A reviewer thought that Figure 4I and its legend are confusing. Particularly, the sentence "Curves become dashed at the point where the observability threshold exceeds 50% of the median cell number" is unclear. The main text does a better job explaining it, but the figure panel and legend can be improved. Can this be clarified, please?

d) The authors refer to the number of "Monte Carlo time steps" in Figure 5 and supplementary information. Can the authors please explain how this is meaningful? Why is this number relevant? A reviewer writes, "The actual kinetic parameter values can have implications on the system behavior when stochasticity is considered, and these parameters being congruent with the expected cell cycle length is relevant. Nevertheless, the relevance of the Monte Carlo time steps on the simulations isn't clear to me."

e) The authors say "repeated the same analysis using annulus sizes of 150 µm, differences in the sizes of neighbors of small and large nests were not seen." Can you please include these results in the supplementary material?

---

## [Author Response]

[…] All issues raised by the reviewers can be considered minor, and are listed below.1) About the concept behind the paper:The reviewers highlight two conceptual questions that would be in need of an answer:a) What are the suggested mechanisms leading to growth arrest of the first cells in a clone? If this were due to some probabilistic event that, however, still depended on oncogene activation, would that not favor the hybrid model B (at least for these initial cells) rather than the authors' preferred model C?

It is true that, for a mechanism in which growth arrest is promoted by a signal from already-arrested cells, something must account for the arrest of the very first cell (or cells). However, that event need not be related to oncogene activation. In our model, as in most models of tissue growth control, dividing cells are simply assumed to have some pre-existing, small probability of spontaneously arresting. Over a fairly wide range of that probability there is little effect on the final size at which clones achieve complete arrest (Kunche et al., 2016)

b) While the model is clearly exclusively or nearly exclusively cell non-autonomous, it is still nevus-autonomous. The authors argue that the OIS model is mostly based on in vitro observations, but should a nevus-autonomous model not also be reproducible in vitro? If it were not, then one would have to invoke an influence of the nevus microenvironment, which would leave room for, among other factors, a senescence-associated program generated by the surrounding cells.

Autonomous collective nevus growth arrest should, in principle, be reproducible in vitro, but in tissue culture many conditions are usually very different from what occurs in vivo. For example, the amount of extracellular volume into which secreted factors are diluted in vitro is typically orders of magnitude more than what a cell might encounter in vivo. Moreover, it is well known that dimensionality (2D versus 3D) has a very large effect on cell behaviors. These factors would make it difficult to interpret any negative in vitro result. Regardless, we strongly agree with the reviewer that the mechanism of growth arrest need not be nevus-autonomous. Cells in the microenvironment could very well have an essential role in relaying or amplifying signals that originate with melanocytes. We do not believe that statements in this manuscript conflict with that possibility. Yet, we would still argue against terming such a mechanism “senescence-associated”, given the evidence we present that arrested nevus cells are not senescent.

What would be the authors' opinion on these two points?2) Regarding the single-cell RNAseq data:a) The authors show data to indicate that BRAF-expressing nevi are no-proliferating, compared to WT hair follicle melanocytes. What about BRAF-expressing melanocytes in hair follicles? Does the system permit an analysis?

Indeed, the cluster we identify as “hair follicle melanocytes” (cluster 3) contains both cells from wildtype (untreated) and mutant (Braf-activated) mice, so a comparison of gene expression is possible. We have added this (using the metaPCNA signature to evaluate proliferation) as a new supplementary figure (Figure 3—figure supplement 2). With the caveat that, as we split up cell groups into smaller subsets we lose statistical power, we failed to observe any statistically significant difference in gene expression between wildtype and mutant cells of this cluster. There was a small trend toward lower proliferative gene expression in the mutant group, so perhaps with more data such an effect might become significant. Regardless, the data show that Braf activation does not drive growth arrest in every context, consistent with the view that cell-cell interactions, rather than Braf itself, are the direct cause of growth arrest.

b) The authors compare their single cell RNA-seq data to many different senescence datasets and signatures – but most of these are from fibroblasts, including Hernandez-Segura et al., 2017. However, Pawlikowski et al., 2013, reported gene expression profiling specifically in primary human melanocytes expressing activated BRAF. Can a comparison to the dataset in this paper be made, as it is judged to be better suited?

Pawlikowski et al., compared gene expression (by microarray) between normal and *BRAF^V600E^*-transduced human melanocytes in vitro. They showed that cultured, BRAF-activated melanocytes display classical morphological features of senescence. This gave us the opportunity to compare how these cells differ from normal cultured melanocytes, and how our nevus cells differ from normal in vivo melanocytes. To the extent that we observe shared differences, we could ask whether those differences are related or unrelated to genes thought to be associated with senescence.

A new supplementary figure (Figure 3—figure supplement 3) addresses that question. In it, we use scatter plots in which the statistically significant gene expression changes seen by Pawlikowski (there are 5063 of them that have unambiguous mouse orthologues) are contrasted with the differences we saw between nevus melanocytes and wildtype cells from either cluster 2 (putative melanocyte stem cells) or cluster 3 (hair follicle melanocytes).

Overall (Figure 3—figure supplement 3A-B), we see only a weak correlation in fold-changes when we compare the Pawlikowski et al., data with our nevus-versus-cluster 3 fold-changes (panel B), a correlation that improves slightly if we restrict the data from the present study to genes that passed a significance cutoff of adjusted *p*<0.05. That correlation mainly derives from a common subset of genes that is strongly downregulated in both the Pawlikowski samples and ours. When we examine these genes closely, we find many are proliferation-associated (panel D), and others are *Mitf*-target genes (panel E; interestingly, in the Pawlikowski data, *Mitf* and all of its target genes are coordinately downregulated, whereas in our case *Mitf* itself is not, and only about half of its targets are). The remaining genes are an interesting set (panel C), but do not overlap in any obvious way with gene sets that have been associated with senescence. On the other hand, when we look specifically at genes that others (specifically, Hernandez-Segura et al., 2017) have associated with senescence (panel F), we see that they very nicely characterize the changes (in both directions) in the Pawlikowski data—but not our data—confirming that the gene expression signatures we chose to use in this manuscript really do apply to melanocytes when they are actually senescent. We thank the reviewer for suggesting this informative analysis.

3) Regarding the mathematical model:a) When the authors are exploring the hypothesis of the multiple stages for a cell-autonomous growth arrest (on Figure 4 legend, and in the Results), the authors use "observation thresholds cannot be so high that the observed median is less than twice the threshold" as an argument to reject some models with few stages. Can you please explain what the logic is behind this restriction? What are the implications?

We apologize for the lack of clarity in the legend to Figure 4, and we have revised it to explain this issue better. What we mean is this: whenever making measurements of a single dispersed quantity many times, one notices an observed dynamic range, i.e. how big the average measurements are compared with the smallest. This number contains empirical information about limits of observability. In modeling, we realized we should stop considering theoretical observation thresholds once they required the observed dynamic range to be implausibly narrow. As shown in Figure 4E, in our in vivo measurements, we see nevi as large as 150 µm in radius, a median of 76.8 µm, and can detect them as small as 40 µm in radius. Assuming a quadratic relationship between radius and cell number, that would represent a dynamic range of observation of 3.7-fold between the median and the smallest. In the interests of being conservative, we chose to consider models implausible once they required a dynamic range of less than two-fold. The dashed lines are used to indicate when modeling parameters produce regimes that should be considered implausible.

b) On Figure 4C-D, a reviewer found it difficult to compare the two models. Can you please explain what the intended message is? It might be clearer if the two distributions are overlapped.

As with the above comment, we were trying to make the legend brief, and ended up being unclear. In panel C, the arrest probability was set at the level required to ensure 99% arrest, whereas in D it was set at the level required to ensure 95% arrest (see panel A). The figure legend has been revised to make this clear. The point is to show that the two distributions are very similar, i.e. the observed heavy-tailed shape is general, and not an artifact of choosing a particular value for the arrest probability. Since we are trying to show that the two distributions are very similar, we find that an overlapped graph is not very visually appealing.

A reviewer thought that Figure 4I and its legend are confusing. Particularly, the sentence "Curves become dashed at the point where the observability threshold exceeds 50% of the median cell number" is unclear. The main text does a better job explaining it, but the figure panel and legend can be improved. Can this be clarified, please?

This is essentially the same as comment 3a. Please see the response above.

d) The authors refer to the number of "Monte Carlo time steps" in Figure 5 and supplementary information. Can the authors please explain how this is meaningful? Why is this number relevant? A reviewer writes, "The actual kinetic parameter values can have implications on the system behavior when stochasticity is considered, and these parameters being congruent with the expected cell cycle length is relevant. Nevertheless, the relevance of the Monte Carlo time steps on the simulations isn't clear to me."

In cellular Potts modeling, cells grow and are displaced over time, and everything about them gets repeatedly recalculated –these are the Monte Carlo time steps. Consistent with reality, cell division in our models is probabilistic—cells experience a nonlinearly increasing likelihood of dividing over time and with increasing size. As a result, there is no exact relationship between Monte Carlo step and cell cycle length, only an empirically-observed, approximate one (which we report as 382 time steps per cell cycle). The importance of reporting this value is to show that times associated with cell growth and division are very much longer than the time intervals over which we re-calculate. As in differential equation modeling, the key to guarding against numerical artifacts is to ensure that calculation time steps are short relative to the characteristic times associated with the events being modeled.

e) The authors say "repeated the same analysis using annulus sizes of 150 µm, differences in the sizes of neighbors of small and large nests were not seen." Can you please include these results in the supplementary material?

We are happy to oblige. The data are now presented as Figure 5—figure supplement 1.